# Chain-of-Thoughts for Molecular Understanding

## Abstract

The adaptation of large language models (LLMs) to chemistry has shown promising performance in molecular understanding tasks, such as generating a text description from a molecule. However, proper reasoning based on molecular structural information remains a significant challenge, e.g., even advanced LLMs such as GPT-4o struggle to identify functional groups which are crucial for inferring the molecular property of interest. To address this limitation, we propose StructCoT, a structure-aware chain-of-thought (CoT) that enhances LLMs' understanding of molecular structures by explicitly injecting the key structural features of molecules. Moreover, we introduce two fine-tuning frameworks for adapting the existing LLMs to use our StructCoT. Our experiments demonstrate that incorporating StructCoT with our fine-tuning frameworks leads to consistent improvements in both molecular understanding tasks.

## 1 Introduction

Large language models (LLMs; Touvron et al., 2023; OpenAI & et al., 2024; Raffel et al., 2020) have demonstrated remarkable performance across various tasks. To leverage their strong capabilities in chemistry, several prior works (Edwards et al., 2022; Christofidellis et al., 2023a; Fang et al., 2024; Pei et al., 2023) have proposed chemical LLMs that have shown superior performance in molecular understanding tasks such as molecule captioning (Mol2Text) and text-based molecule generation (Text2Mol) (Edwards et al., 2022), which are crucial for designing new molecules.

Reasoning based on molecular structures plays an important role in molecular understanding tasks in practice. For example, chemists are likely to consider a molecule toxic if it contains a phenol group due to the formation of phenoxyl radicals and the compound's ability to interact with biological membranes (Hansch et al., 2000). However, despite its significance, there exists a lack of studies on the role of reasoning in LLM-based molecular understanding. In other domains such as arithmetic and commonsense reasoning, chain-of-thought (CoT; Wei et al., 2022; Kojima et al., 2022) has shown that explicitly incorporating such reasoning steps significantly improves the performance of LLMs. In detail, CoT aims to generate intermediate reasoning steps before arriving at a final answer.

One might consider the naive adaptation of CoT prompting to include molecular structural information in reasoning. However, we observe this to be ineffective because even state-of-the-art LLMs (OpenAI & et al., 2024; Touvron et al., 2023) struggle to capture the structural details of molecules, as described in Figure 1 and Section 3.2. This hinders their ability to perform reasoning effectively in molecular understanding tasks. While some prior works (Ouyang et al., 2024; Jin et al., 2024; M. Bran et al., 2024) have proposed CoTs for chemistry, they are either not applicable or exhibit limited performance in molecular understanding tasks.

In this paper, we propose StructCoT, a chain-of-thought that progressively sketches the structural features of molecules to solve molecular understanding tasks. StructCoT consists of six key structural elements, ranging from the primary structure to the smaller components. We propose to explicitly inject the appropriate structural information with StructCoT to enhance the language models' understanding of molecules, which compensates for the lack of structural information.

Moreover, we propose two different fine-tuning frameworks to apply StructCoT depending on the input and output of the given molecular understanding task, as illustrated in Figure 2. Both approaches share the same outline, including a reasoning module that generates StructCoT and an

Figure 1: **The failure case of GPT-4o for the inference of structural information given the molecular SMILES.** The red color indicates the wrong generated structural information while the green color indicates the correct answer. Note that we visualize the molecular graph for illustration purposes; GPT-4o does not take them as inputs.

answering module that generates the output using the input combined with STRUCTCoT. On the one hand, for the molecule captioning task, we use external tools like RDKit (Landrum et al., 2024) as the reasoning module, since they can precisely determine the structural information from the molecule. Therefore, one attaches a perfectly accurate STRUCTCoT to the input Simplified Molecular Input Line Entry System (SMILES; Weininger, 1988) and lets the answering module generate the output.

On the other hand, for the text-based molecule generation task, one cannot acquire the exact STRUCTCoT as the molecule is not provided. Therefore, we propose to finetune the LLMs as the reasoning module that generates STRUCTCoT (Ho et al., 2023; Fu et al., 2023a; Magister et al., 2023). Then, we fine-tune the answering module to generate the answer given the text description and the acquired STRUCTCoT. Moreover, we incorporate a novel *matching-ratio-based rejection sampling* into the answering module, which forces the structure of the generated molecule to align with the structural information in STRUCTCoT. Notably, the proposed rejection sampling leverages the deterministic nature of structural information for a given molecule.

As a result, incorporating our proposed method into both chemistry LLMs (Edwards et al., 2022; Christofidellis et al., 2023a) and general LLMs (Touvron et al., 2023; OpenAI & et al., 2024) leads to consistent performance improvements. Specifically, when incorporated with MolT5-large (Edwards et al., 2022) and Text+Chem T5 (Christofidellis et al., 2023a), our method achieves competitive performance with recent baselines in both tasks. In summary, our key contributions are as follows:

- We present the limitations of LLMs in understanding molecular structures by analyzing their capability to infer structural information.
- We introduce STRUCTCoT, a chain-of-thought that progressively sketches the structural information of molecules, for the reasoning of molecular understanding.
- We design a framework to incorporate STRUCTCoT for molecule captioning by fine-tuning the answering module with the deterministic and perfectly accurate STRUCTCoT.
- We design a framework to incorporate STRUCTCoT for text-based molecule generation by applying CoT fine-tuning for the reasoning module, fine-tuning the answering module, and a novel matching ratio-based rejection sampling which further improves the performance.
- We validate the efficacy of STRUCTCoT and our fine-tuning framework by showing consistent improvements across chemical and general LLMs.

## 2 RELATED WORK

**Large language models for chemistry.** General-purpose large language models (generalist LLMs) often struggle to solve basic chemistry problems and molecular understanding tasks (White et al.,

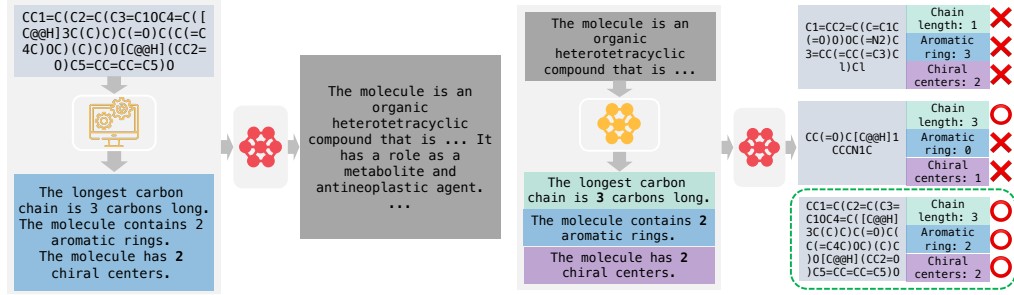

(a) Molecule captioning (Mol2Text)          (b) Text-based molecule generation (Text2Mol)

Figure 2: **Overview of the Fine-tuning Framework of STRUCTCOT**. Light gray boxes represent SMILES strings; gray boxes represent text descriptions; colored boxes represent STRUCTCOT. The yellow ones are the reasoning module, and the red ones are the answering module. In (b), colors indicate each STRUCTCOT and the corresponding structural information elements. The third SMILES is selected after matching ratio-based rejection sampling for having the highest matching ratio (3/3).

2023; Castro Nascimento & Pimentel, 2023; Guo et al., 2023). To address this issue, prior works have introduced specialist LLMs, i.e., chemical LLMs, by pre-training models on molecule-related texts (Edwards et al., 2022; Christofidellis et al., 2023b; Liu et al., 2023a; Pei et al., 2023), through instruction tuning (Fang et al., 2024; Cao et al., 2023), and using retrieval-based in-context learning (Li et al., 2024a). Additionally, some works have improved LLMs by incorporating graph or 3D coordinate information (Liu et al., 2023b; Li et al., 2024b; Liu et al., 2024). Our work focuses on reasoning processes that are broadly applicable to these specialist LLMs as well as generalist LLMs.

**Chain-of-thought reasoning.** Chain-of-thought (CoT) aims to generate intermediate reasoning steps before arriving at a final answer (Wei et al., 2022; Kojima et al., 2022). CoT not only enhances the reasoning capabilities of LLMs but also improves the overall quality of generated answers. Most prior works generated CoTs via few-shot learning based on the manually written CoTs (Wei et al., 2022) or by prompting LLMs with "Let's think step by step." (Kojima et al., 2022). In addition, several approaches have proposed to further enhance CoT, including techniques such as self-consistency (Wang et al., 2023), least-to-most prompting (Zhou et al., 2023), complexity-based prompting (Fu et al., 2023b), and self-polish (Xi et al., 2023). However, the ability to perform complex reasoning remains limited to extremely large language models (>100B parameters).

To address this challenge, various approaches have been introduced to distill knowledge from very large language models to smaller ones (<10B). Specifically, Ho et al. (2023); Fu et al. (2023a); Magister et al. (2023) employed the larger models as teacher models to generate CoTs for fine-tuning smaller student models. Nevertheless, even recent LLMs struggle to generate appropriate CoTs that demonstrate a correct understanding of molecular structures (as described in Figure 1 and Section 3.2), restricting the efficacy of LLMs in generating appropriate CoTs for molecular understanding tasks.

**Chain-of-thought reasoning for chemistry.** Recently, a few works have extended CoT reasoning to address chemistry-related problems. For instance, Ouyang et al. (2024) proposed to employ the program-of-thoughts (PoT; Chen et al., 2023) to handle chemical question-answering tasks. Additionally, Jin et al. (2024) presented the protein chain of thought (ProCoT) to replicate the signaling pathways in the context of the protein-protein interaction (PPI) problem. Despite these advances, none of these works target molecular understanding tasks such as molecule captioning and text-based molecule generation. We note that M. Bran et al. (2024) provided CoT comparable to ours, but their CoTs are less focused on molecule structural reasoning, e.g., they propose CoTs based on tools like *LitSearch/WebSearch*, *PatentCheck*, *ReactionPlanner*, and *SMILES2Price*. Moreover, it shows limited performance improvements in molecule understanding tasks as observed in **??**.

## 3    STRUCTURE AS MILESTONES OF LLM-BASED CHEMICAL REASONING

In this section, we emphasize the importance of incorporating molecular structural information into the reasoning of LLMs for molecular understanding. We first outline key structural information essential for understanding the chemical and physical properties of a molecule, providing specific

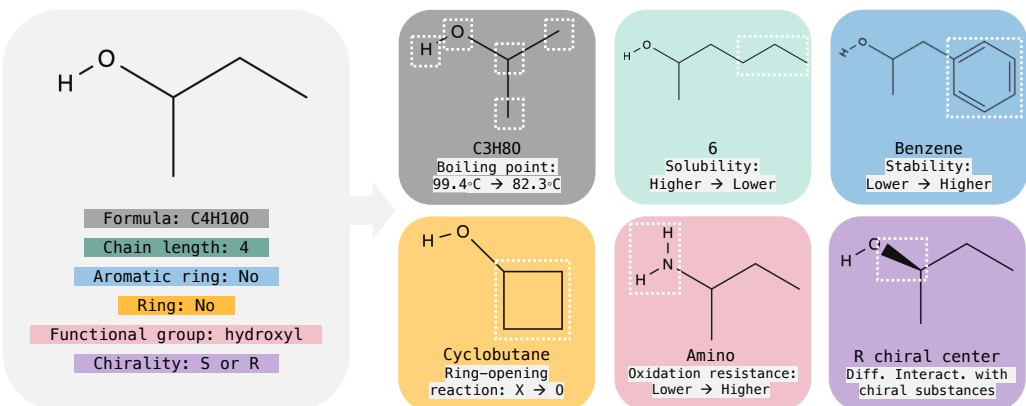

Figure 3: **Illustration of the Importance of Structural Information.** This illustrates an example of replacing each structural information (described with a dashed box) of the molecule. From left to right and top to bottom, the described structural information are molecular formula, longest carbon chain length, aromatic ring, ring compounds, functional group, and chirality.

examples. Then, we show that even the state-of-the-art LLMs, such as GPT-4o (OpenAI & et al., 2024) and Llama3-8B-Instruct (Touvron et al., 2023), often fail to accurately infer crucial structural details from the molecule or the text description of the molecule. This observation implies that recent LLMs may struggle to implicitly reason these foundational structural elements when tackling molecular tasks, highlighting the potential benefits of explicitly integrating such information through a chain-of-thought approach.

### 3.1 EXAMPLES OF IMPORTANT STRUCTURAL INFORMATION

Humans typically analyze a molecule by progressively mapping its structure, starting with primary elements like rings and long carbon chains, and then identifying smaller components such as functional groups and chiral centers. Reflecting this approach, we identify six key elements of molecular structure that are critical for chemical reasoning. To highlight the importance of these structural elements, we demonstrate how even slight modifications in molecular structure can lead to significant changes in chemical or physical properties, as shown in Figure 3.

**Molecular formula.** The molecular formula provides essential information about a molecule's composition, specifying the number and type of atoms present. This information is critical because, for example, it directly determines the molecular weight. To illustrate, although 2-Butanol ($C_4H_{10}O$) and 2-Propanol ($C_3H_8O$) are composed of the same type of atoms, i.e., carbon, hydrogen, and oxygen, their differing molecular formulas result in distinct molecular weights (74.1g/mol for 2-Butanol and 60.1g/mol for 2-Propanol). These differences lead to the change in boiling points, $99.4°C$ and $82.3°C$, respectively, as shown in the gray part of Figure 3.

**Longest carbon chain.** The longest carbon chain (excluding atoms in ring systems) forms the molecular backbone where functional groups are attached. The length of this chain significantly influences properties like solubility. For example, extending the carbon chain of 2-Butanol from four to six carbons creates 2-Hexanol, which exhibits reduced solubility. This is illustrated in the green section of Figure 3.

**Aromatic rings.** Aromatic rings, such as benzene or pyridine, play a critical role in determining the stability and electronic properties of molecules. For instance, adding a benzene ring to 2-Butanol yields 1-Phenyl-2-Propanol, which has enhanced stability and greater oxidation resistance. This transformation is shown in the blue section of Figure 3.

**Ring compounds.** Similar to the longest carbon chain, ring structures often serve as the backbone where functional groups are attached. The ring system significantly affects molecular behavior and reactions. For example, although 2-Butanol and Cyclobutanol share the same number of carbons and oxygen, the ring in Cyclobutanol introduces a tendency toward ring-opening reactions, as depicted in the yellow section of Figure 3.

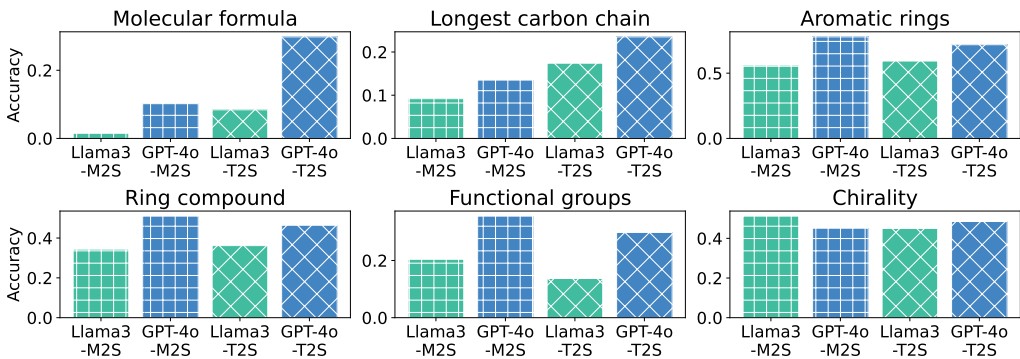

Figure 4: **Analysis of LLMs' Understanding of Structural Information.** Colors indicate the architectures of the language models, with green and blue representing LLaMA3-8B and GPT-4, respectively. Patterns denote the input types: crossed patterns represent SMILES representations (*Molecule2Structure*), and diagonally crossed patterns represent molecule captions (*Text2Structure*).

**Functional groups.** Functional groups, e.g., hydroxyl, amino, ester, etc., play a pivotal role in determining the chemical reactivity of molecules. For example, alcohols with a hydroxyl group (-OH) are prone to oxidize more while the molecules with an amino group (-NH$_2$) are generally resistant to oxidation under mild conditions. A single replacement of a hydroxyl (-OH) group in 2-Butanol with an amino (-NH$_2$) group leads to 2-Butanamine, which has increased oxidation resistance, as described in the red part of Figure 3.

**Chiral centers.** Chirality refers to the stereochemical property of a molecule that makes it non-superimposable on its mirror image, leading to different chemical behaviors. The chirality is determined by the chiral centers and their configurations, i.e., R- and S-configuration [1], which describe the spatial arrangement of the groups around the chiral centers. This leads to different interactions between other molecules with chirality. For instance, (R)-2-Butanol and (S)-2-Butanol may interact differently with other chiral substances. This is described in the purple part of Figure 3.

### 3.2 RECENT LARGE LANGUAGE MODELS DO NOT UNDERSTAND STRUCTURAL INFORMATION

Next, we demonstrate that even recent LLMs, i.e., GPT-4o (OpenAI & et al., 2024) and LlaMA3-8B-Instruct (Touvron et al., 2023), fail to infer important structural information from the given molecule and the text description of the molecule. We evaluate the LLMs by querying the structural information from the SMILES string (Weininger, 1988) and the text description, which can be considered as a simple task that could be solved by someone with a bachelor's degree in chemistry.

As shown in Figure 4, both GPT-4o and LlaMA3-8B-Instruct fail to capture the structural information accurately. First, when the SMILES string is provided, both models perform best in counting the number of aromatic rings, with accuracies around 50% and 75%, respectively. However, their accuracies are significantly lower for other structural information. This implies that LLMs cannot fully understand the molecular structures given the molecular string. We provide an example of a failure case in Figure 1.

Similarly, when the text description is given, both models also fail to achieve a high accuracy in inferring the structural information. This indicates that LLMs cannot properly understand the structure of molecules even when provided with the text description of molecules. These observation highlight the potential benefits of explicilty incorporating structural CoT to enhance molecular comprehension. Note that we provide the detailed experimental settings and prompts for the analysis in Appendix A.1.

## 4 STRUCTCoT: STRUCTURE-AWARE CoTS FOR MOLECULES

In this section, we describe our framework to enhance the capability of language models to perform reasoning using structure-aware CoTs of molecules (STRUCTCoT). Although our method is broadly applicable, we focus on two tasks commonly used to evaluate the ability of LLMs to understand

---

[1]The names of R and S come from the Latin word *Rectus* and *Sinister*, which means right and left, respectively.

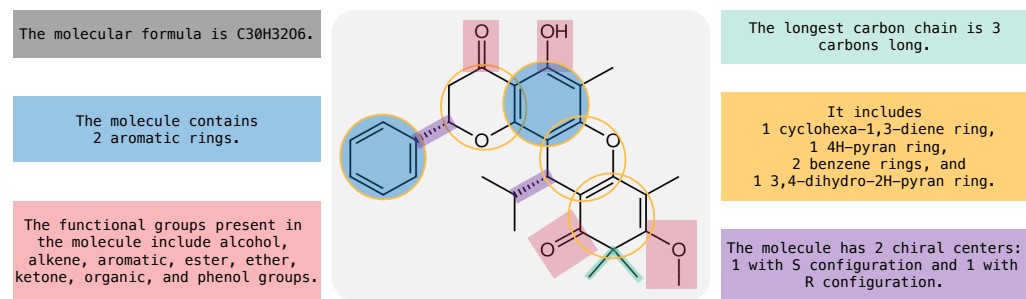

Figure 5: **The Six New Elements of STRUCTCOT: molecular formula, longest carbon chain length, aromatic rings, ring compounds, functional groups, and chirality.** The same color indicates the CoT and the corresponding structural information of the molecule. The order of the STRUCTCOT follows the order mentioned in the title of the figure, which progressively sketches the structure of molecules.

chemical knowledge (Edwards et al., 2022). The first task is molecule captioning (Mol2Text), where the goal is to generate a text description from an input molecule's SMILES representation. The second task is text-based molecule generation (Text2Mol), where the LLM aims to generate a molecule that corresponds to a given textual description.

We incorporate our STRUCTCOT through a two-stage procedure of reasoning and answering. In the reasoning step, a *reasoning module* generates STRUCTCOT that will be used as additional structural information for understanding the molecule. Next, in the answering step, an *answering module* generates the answer from the input augmented with the generated CoTs. Note that we separate the two different architectures for each task since the reasoning module differs by the task: (1) one has access to the ground-truth reasoner for molecule captioning and (2) one needs to additionally fine-tune the reasoning module for improved reasoning capability for text-based molecule generation.

The rest of this section is organized as follows. First, in Section 4.1, we introduce STRUCTCOT, the structure-aware CoTs inspired by the significance of structural information explained in Section 3.1. Then, in Section 4.2 and Section 4.3, we present the fine-tuning process to incorporate the STRUCTCOT into both molecule captioning and text-based molecule generation tasks.

## 4.1 STRUCTCOT

We introduce STRUCTCOT, a structure-aware CoT designed to enhance language models' understanding of molecular structures. Each component of STRUCTCOT follows the six important structural information introduced in Section 3.1 and illustrated in Figure 5.

**Molecular formula** is expressed as *"The molecular formula is $X_1N_1 \cdots X_MN_M$."*, where $X_m$ and $N_m$ represent the $m$-th atom type and the associated number of atoms, respectively.

**Length of the longest carbon chain** takes the following form: *"The longest carbon chain length is $N$ carbons long."*, where $N$ denotes the length of the longest carbon chain of the molecule.

**Number of aromatic rings** takes the following form: *"The molecule contains $X$ aromatic ring(s)."*, where $X$ denotes the number of aromatic rings in the molecule.

**Types of ring compounds** is expressed as *"It includes $N_1 X_1$ rings, $\cdots$, $N_M X_M$ ring(s)."*, where $X_m, N_m$ represents the International Union of Pure Applied Chemistry (IUPAC) name of the ring compound and the number of the rings, respectively.

**Types of functional groups** is expressed as *"The functional groups present in the molecule include $X_1, X_2, \cdots$, and $X_N$ group."*, where $X_n$ denotes the name of the functional group.[2]

**Number and types of chiral centers** is formulated as follows: *"The molecule has $N$ chiral centers: $N_S$ with S configuration and $N_R$ with R configuration."*, where $N = N_S + N_R$, and $N_S$ and $N_R$ denotes the number of chiral centers of S and R configurations, respectively.

---

[2]Note that we consider a wider range of functional groups compared to that of M. Bran et al. (2024).

## 4.2 MOLECULE CAPTIONING

Molecule captioning aims to generate an accurate and detailed text description of a given molecular SMILES string. We incorporate our STRUCTCOT scheme through (1) using external tools like RDKit (Landrum et al., 2024) as a ground-truth reasoning module and (2) fine-tuning the answering module LLM with the generated CoT as an additional input. We provide the description in Figure 2a.

**Reasoning module.** One can obtain the true structural information of the given molecule from RDKit, which allows us to guide the answering module without uncertainty. This is natural as the structural information is deterministic given the molecule. Consequently, the obtained true structural information is used as STRUCTCOT. For this task, we consider the molecular weight CoT and IUPAC name CoTs (M. Bran et al., 2024) in addition to the CoTs described in Section 4.1.

**Answering module.** With the molecule and the acquired CoT as an input, we fine-tune the LLMs to generate the description of the molecule. In the experiments, we mainly consider chemical LLMs, i.e., MolT5 (Edwards et al., 2022) and ChemT5 (Christofidellis et al., 2023a), as the answering module.

## 4.3 TEXT-BASED MOLECULE GENERATION

Text-based molecule generation is the reverse process of molecule captioning, intending to generate the corresponding molecular string based on the given description. Following the two-stage framework that separates rationale generation and answer inference (Zhang et al., 2024), we first generate STRUCTCOT using the fine-tuned reasoning module and then attach this to the input and employ this as an input for the answering module. We provide the description in Figure 2b.

Notably, we selectively use CoT elements in STRUCTCOT. This is because the reasoning modules need to generate CoTs of sufficient quality for the answering module, but this is not possible for some types of CoTs. Therefore, we evaluate the abilities of the reasoning module to correctly generate the CoTs and exclude those with low accuracy (presented in Table 2), specifically the molecular formula CoT and the two CoTs proposed by M. Bran et al. (2024).

**Reasoning module.** For the reasoning module, following Ho et al. (2023); Fu et al. (2023a); Magister et al. (2023), we enable CoT reasoning of the models by fine-tuning the reasoning module on the STRUCTCOT as the molecule is not given. This is in contrast to the molecule captioning task where the exact structural information can be extracted from external tools with the given molecule. We mainly fine-tune the chemical LLMs, i.e., MolT5 and ChemT5 for this module.

**Answering module.** For the answering module, similar to that of molecule captioning, we fine-tune a chemical LLM to generate an appropriate molecule given the text description and generated STRUCTCOT. Moreover, we propose the *matching ratio-based rejection sampling*, which forces the generated molecule to align with STRUCTCOT, as described in the following.

The proposed matching ratio-based rejection sampling aims to match the structural information of the generated molecule with the given STRUCTCOT. In detail, we generate multiple $k$ molecules using beam search and then score each molecule based on the matching ratio, which counts the number of matching structural information elements between STRUCTCOT and the generated molecule. Finally, we choose the best-scoring molecule as the final output. This approach also leverages the deterministic nature of structural information, i.e., we can easily compare the alignment between each structural information and the generated molecule. Notably, this differs from the prior works with iterative approaches (Wang et al., 2023; Xi et al., 2023; Sun et al., 2024), as we focus on the alignment between CoT and generated answer without needing to generate multiple rationales.

## 5 EXPERIMENTS

In this section, we present our experiments on molecule captioning and text-based molecule generation tasks, including the experimental results, setting details, and ablation studies. We first explain the common settings shared by both tasks. Note that we provide the experimental results on a retrosynthesis task in Appendix B.4.

**Dataset.** Following prior works (Edwards et al., 2022; Christofidellis et al., 2023a), we employ the widely used CHEBI-20 dataset (Edwards et al., 2021), which consists of 33,010 pairs of molecular

Table 1: **Molecule Captioning Performance.** Δ denotes the performance difference between the original model and the one incorporated with STRUCTCOT. Teal color indicates the improvement.

| Models | BLEU-2 Metric | Δ | BLEU-4 Metric | Δ | ROUGE-1 Metric | Δ | ROUGE-2 Metric | Δ | ROUGE-L Metric | Δ | METEOR Metric | Δ |
|---|---|---|---|---|---|---|---|---|---|---|---|---|
| *Baselines (without CoTs)* | | | | | | | | | | | | |
| RNN | 0.251 | - | 0.176 | - | 0.450 | - | 0.278 | - | 0.394 | - | 0.363 | - |
| T5-base | 0.511 | - | 0.423 | - | 0.607 | - | 0.451 | - | 0.550 | - | 0.539 | - |
| Transformer | 0.061 | - | 0.027 | - | 0.204 | - | 0.087 | - | 0.186 | - | 0.114 | - |
| MolXPT | 0.594 | - | 0.505 | - | 0.660 | - | 0.511 | - | 0.597 | - | 0.626 | - |
| BioT5 | 0.635 | - | 0.556 | - | 0.692 | - | 0.559 | - | 0.633 | - | 0.656 | - |
| *Specialists (fine-tuning)* | | | | | | | | | | | | |
| MolT5-base | 0.540 | - | 0.457 | - | 0.634 | - | 0.485 | - | 0.578 | - | 0.569 | - |
| +STRUCTCOT | 0.592 | 0.052 | 0.507 | 0.050 | 0.667 | 0.043 | 0.523 | 0.038 | 0.606 | 0.028 | 0.619 | 0.050 |
| MolT5-large | 0.594 | - | 0.508 | - | 0.654 | - | 0.510 | - | 0.594 | - | 0.614 | - |
| +STRUCTCOT | **0.645** | 0.051 | **0.567** | 0.059 | **0.699** | 0.045 | **0.568** | 0.058 | **0.639** | 0.045 | **0.666** | 0.052 |
| ChemT5-small | 0.553 | - | 0.462 | - | 0.633 | - | 0.481 | - | 0.574 | - | 0.583 | - |
| +STRUCTCOT | 0.601 | 0.048 | 0.513 | 0.050 | 0.664 | 0.031 | 0.519 | 0.038 | 0.603 | 0.029 | 0.624 | 0.042 |
| ChemT5-base | 0.580 | - | 0.490 | - | 0.647 | - | 0.498 | - | 0.586 | - | 0.604 | - |
| +STRUCTCOT | 0.639 | 0.059 | 0.560 | 0.070 | 0.687 | 0.040 | 0.553 | 0.055 | 0.626 | 0.040 | 0.657 | 0.053 |
| *Generalists (without fine-tuning)* | | | | | | | | | | | | |
| Llama3 | 0.211 | - | 0.117 | - | 0.367 | - | 0.183 | - | 0.308 | - | 0.257 | - |
| +STRUCTCOT | 0.259 | 0.048 | 0.158 | 0.041 | 0.401 | 0.034 | 0.208 | 0.025 | 0.324 | 0.016 | 0.341 | 0.084 |
| GPT-4o | 0.232 | - | 0.128 | - | 0.389 | - | 0.183 | - | 0.307 | - | 0.291 | - |
| +STRUCTCOT | 0.286 | 0.054 | 0.174 | 0.046 | 0.405 | 0.016 | 0.199 | 0.016 | 0.313 | 0.006 | 0.341 | 0.050 |
| Mol-Instructions | 0.217 | - | 0.143 | - | 0.337 | - | 0.196 | - | 0.291 | - | 0.254 | - |
| +STRUCTCOT | 0.347 | 0.130 | 0.275 | 0.132 | 0.601 | 0.264 | 0.518 | 0.322 | 0.593 | 0.302 | 0.520 | 0.266 |

SMILES and their text descriptions. We also use the same train/validation/test split of 80%/10%/10%. Note that for experiments involving Mol-Instructions (Fang et al., 2024), we used the Mol-Instructions dataset as provided in their work.

**Baselines.** We verify the performance enhancement of STRUCTCOT in two settings: specialist and generalist models. On the one hand, we employed two popular specialist models, i.e., chemical LLMs: MolT5 (Edwards et al., 2022) and Text+CHem T5 (ChemT5; Christofidellis et al., 2023a). To validate the efficacy of our method across various model sizes, we used small (77M) and base (252M) for ChemT5 and base and large (800M) for MolT5. On the other hand, we employed three recent large language models: Llama3-8B-Instruct (Touvron et al., 2023), GPT-4o (OpenAI & et al., 2024)[3], and Mol-Instructions (Fang et al., 2024) as our generalist models. Additionally, we include five baselines including RNN (Jain & Medsker, 1999), Transformer (Vaswani et al., 2017), T5 (Raffel et al., 2020), MolXPT (Liu et al., 2023a), and BioT5 (Pei et al., 2023) to compare the absolute performance.

## 5.1 MOLECULE CAPTIONING

**Experimental setup and metrics.** For specialist models, we follow the method proposed in Section 4.2. For the generalist models without any domain-specific instruction tuning (Llama3 and GPT-4o), we cannot guarantee that the generated descriptions align with those in our training data. Therefore, we apply 10-shot learning by attaching CoTs in the same way as for the specialist models. Lastly, for Mol-Instructions, we do not apply fine-tuning or few-shot learning but prompt the model with instructions enriched with CoTs. Performance is evaluated by comparing the generated captions with the ground-truth captions using six metrics: BLEU-2, BLEU-4 (Papineni et al., 2002), ROUGE-1, ROUGE-2, ROUGE-

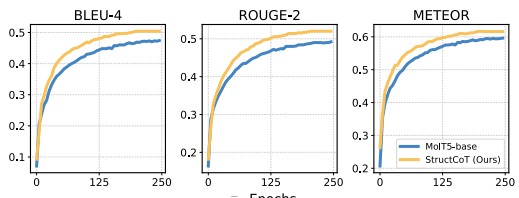

Figure 6: **Comparison of with and without STRUCTCOT (MolT5-base).** Incorporating STRUCTCOT improved the performance faster.

---

[3]We used `gpt-4o-2024-05-13`.

Table 2: **Reasoning Accuracy for Each Structural Information.**

| Models | Form. | Chain | Arom. | Ring | Func. | Chiral. | Weight | Name |
|---|---|---|---|---|---|---|---|---|
| *Specialists (fine-tuning)* | | | | | | | | |
| MolT5-base | 0.458 | 0.922 | 0.926 | 0.930 | 0.957 | 0.798 | 0.606 | 0.512 |
| ChemT5-small | 0.447 | 0.920 | 0.930 | 0.926 | 0.954 | 0.788 | 0.634 | 0.495 |
| ChemT5-base | 0.475 | 0.925 | 0.931 | 0.930 | 0.960 | 0.799 | 0.641 | 0.525 |
| *Generalists* | | | | | | | | |
| Llama3 | 0.084 | 0.174 | 0.593 | 0.362 | 0.137 | 0.450 | 0.435 | 0.015 |
| GPT-4o | 0.298 | 0.235 | 0.718 | 0.464 | 0.298 | 0.485 | 0.728 | 0.040 |

Table 3: **Text-based Molecule Generation Performance.** The teal color indicates the improvement while the red color indicates the reduction.

| Models | BLEU Met. | Δ | Exact Met. | Δ | Levenshtein↓ Met. | Δ | MACCS FTS Met. | Δ | RDK FTS Met. | Δ | Morgan FTS Met. | Δ | FCD↓ Met. | Δ | Validity Met. | Δ |
|---|---|---|---|---|---|---|---|---|---|---|---|---|---|---|---|---|
| *Baselines (without CoTs)* | | | | | | | | | | | | | | | | |
| RNN | 0.652 | - | 0.005 | - | 38.09 | - | 0.591 | - | 0.400 | - | 0.362 | - | 4.55 | - | 0.542 | - |
| Transformer | 0.499 | - | 0.000 | - | 57.66 | - | 0.480 | - | 0.320 | - | 0.217 | - | 11.32 | - | 0.906 | - |
| T5-base | 0.762 | - | 0.069 | - | 24.95 | - | 0.731 | - | 0.605 | - | 0.545 | - | 2.48 | - | 0.660 | - |
| MolXPT | - | - | 0.215 | - | - | - | 0.859 | - | 0.757 | - | 0.667 | - | 0.45 | - | 0.983 | |
| BioT5 | 0.867 | - | 0.413 | - | 15.10 | - | 0.886 | - | 0.801 | - | 0.734 | - | 0.43 | - | **1.000** | |
| *Specialists (fine-tuning)* | | | | | | | | | | | | | | | | |
| MolT5-base | 0.769 | - | 0.081 | - | 24.46 | - | 0.721 | - | 0.588 | - | 0.529 | - | 2.18 | - | 0.772 | - |
| +STRUCTCOT | 0.863 | 0.094 | 0.385 | 0.304 | 13.91 | 10.55 | 0.918 | 0.197 | 0.843 | 0.255 | 0.783 | 0.254 | 0.29 | 1.89 | 0.983 | 0.211 |
| MolT5-large | 0.854 | - | 0.311 | - | 16.07 | - | 0.834 | - | 0.746 | - | 0.684 | - | 1.20 | - | 0.905 | - |
| +STRUCTCOT | **0.886** | 0.032 | 0.391 | 0.080 | 12.98 | 3.09 | 0.906 | 0.072 | 0.822 | 0.076 | 0.765 | 0.081 | 0.35 | 0.085 | 0.947 | 0.042 |
| ChemT5-small | 0.739 | - | 0.157 | - | 28.54 | - | 0.859 | - | 0.736 | - | 0.660 | - | 0.07 | - | 0.776 | - |
| +STRUCTCOT | 0.874 | 0.135 | 0.381 | 0.224 | 13.22 | 15.32 | 0.918 | 0.059 | 0.845 | 0.109 | 0.787 | 0.127 | 0.29 | 0.22 | 0.976 | 0.200 |
| ChemT5-base | 0.750 | - | 0.212 | - | 27.39 | - | 0.874 | - | 0.767 | - | 0.697 | - | **0.06** | - | 0.792 | - |
| +STRUCTCOT | 0.878 | 0.128 | **0.421** | 0.209 | **12.76** | 14.63 | **0.924** | 0.050 | **0.856** | 0.089 | **0.804** | 0.107 | 0.26 | 0.20 | 0.982 | 0.190 |

L (Banerjee & Lavie, 2005), and METEOR (Banerjee & Lavie, 2005). We provide detailed experimental settings and prompts in Appendix A.2.

**Results.** We report the experimental results in Table 1. We observe that adding STRUCTCOT consistently improves performance for both specialist and generalist models. Surprisingly, despite BioT5 being pre-trained on a larger dataset and sharing the same model size, our method, when incorporated with ChemT5-base, achieves competitive results without any additional pre-training data. We provide an example generated sample in Figure 7 and more examples in Appendix B.1. Moreover, our approach shows faster performance improvement, as illustrated in Figure 6.

## 5.2 TEXT-BASED MOLECULE GENERATION

**Experimental setup and metrics.** We follow the fine-tuning framework proposed in Section 4.3. The performance is evaluated by comparing the generated molecules with the reference molecules using eight metrics: SMILES comparison metrics (BLEU, Exact, and Levenshtein distance (Miller et al., 2009)), fingerprint similarity metrics (MACCS FTS (Durant et al., 2002), RDK FTS (Schneider et al., 2015), and Morgan FTS (Rogers & Hahn, 2010)), a molecular distribution metric (Fréchet ChemNet Distance (FCD) (Preuer et al., 2018)), and the validity of the generated molecule. We provide detailed experimental settings and prompts in Appendix A.3. Notably, we do not report the performance of generalist models in the main text because their reasoning accuracy is very low, as shown in Table 2. This low accuracy implies that their reasoning would not guide the answer appropriately, even in the few-shot learning setting. However, we include these results in Appendix B.2 for completeness. We share the model weights for the reasoning and the answering modules when experimenting on the MolT5-large, since it leads to slightly better performance.

**Reasoning accuracy.** We first measure the reasoning accuracy to filter out low-accuracy reasoning components that may misguide the answer module. Specifically, the accuracies for molecular formula, longest carbon chain length, number of aromatic rings, chirality, and IUPAC names are computed by exact match. The accuracies for ring compounds and functional groups are computed by the ratio of

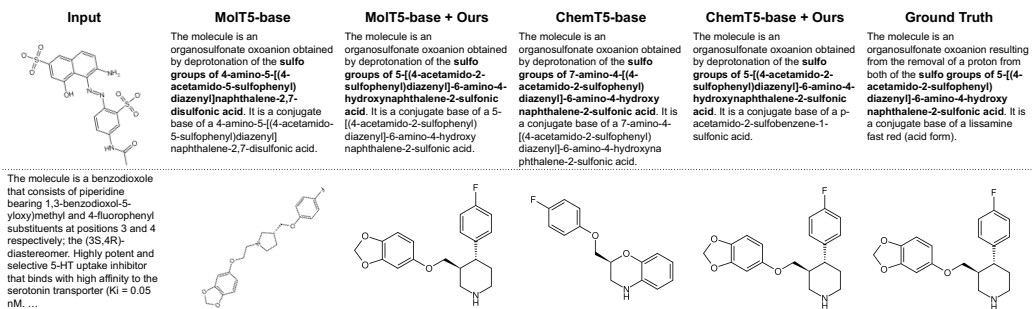

Figure 7: **Examples of generated samples.** A Mol2Text sample is at the top and a Text2Mol sample is at the bottom. We provide more examples in Appendix B.1 and Appendix B.2.

intersection between the set of true and generated CoTs. Lastly, the accuracy for molecular weight is considered correct if the generated weight is within 95% to 105% of the true weight.

The reasoning accuracies are provided in Table 2. Our results show that our fine-tuned specialist reasoning modules exhibit superior reasoning accuracy compared to larger generalist models, underscoring their ability to understand molecular structures effectively. However, even our reasoning modules failed to achieve high accuracy in molecular formula, molecular weight, and IUPAC name. Therefore, we filter out these three STRUCTCOT components.

**Results.** The experimental results are reported in Table 3. Incorporating our generated STRUCTCOT to the molecular description always improved performance. In particular, incorporating STRUCTCOT into the ChemT5-base achieves state-of-the-art performance compared to the recent baselines, validating the efficacy of our CoTs. Surprisingly, our STRUCTCOT even improves the performance of smaller models beyond that of the vanilla larger models, e.g., MolT5-base+STRUCTCOT showed superior performance to MolT5-large. We provide an example generated sample in Figure 7 and more examples in Appendix B.1.

### 5.3 ABLATION STUDY

We conduct ablation studies on matching ratio-based rejection sampling and each structural component, as well as an experiment comparing our method to ChemCrow. Due to limited space, we present the results of the structural component analysis and the comparison to ChemCrow in Appendix B.3.

**Matching ratio-based rejection sampling.** Here, we discuss the efficacy of matching ratio-based rejection sampling and the impact of the number of samples $k$ in text-based molecule generation. We compare the results of ChemT5-small, both without ($k = 1$) and with the rejection sampling ($k \in \{5, 10, 20\}$). As demonstrated in Figure 8, the rejection sampling improves performance by encouraging the output to follow the given STRUCTCOT. Notably, increasing $k$ beyond 5 does not further improve performance, implying that a choice of $k = 5$ is sufficient.

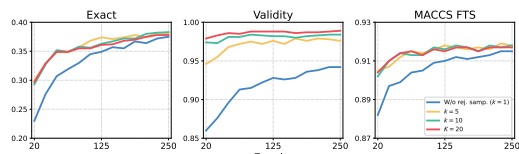

Figure 8: **Ablation study on $k$ of matching Ratio-based Rejection Sampling.**

## 6 CONCLUSION

In this paper, we introduced STRUCTCOT, a structure-aware chain-of-thought framework that enhances language models' understanding of molecular structures by explicitly incorporating key structural features. Our analysis demonstrated that recent large language models struggle to accurately infer structural information from molecular representations like SMILES strings or textual descriptions, highlighting the need for explicit structural reasoning. By fine-tuning domain-specific specialist models with STRUCTCOT, we achieved consistent improvements in molecule captioning and text-based molecule generation tasks. This work underscores the effectiveness of small, domain-specific models in capturing molecular structures, and offers a solution for molecular reasoning.

REPRODUCIBILITY

All experimental code related to this paper is available at https://anonymous.4open.science/r/MolStructCoT. Detailed insights regarding the experiments, encompassing dataset and model specifics, are available in Section 5. For intricate details like hyperparameters, consult Appendix A.

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

# A EXPERIMENTAL DETAILS

In this section, we provide the details of the experiments. All experimental code related to this paper is available at https://anonymous.4open.science/r/MolStructCoT.

## A.1 STRUCTURE INFORMATION ANALYSIS

Here, we describe the detailed settings for the analysis in Section 3.1. To evaluate the understanding of two recent LLMs: Llama3-8B-Instruct (Touvron et al., 2023) and GPT-4o (OpenAI & et al., 2024), we prompt the LLMs to infer the structural information from the given molecular SMILES string and text description of the molecule.

**Prompts given SMILES string.** First, we asked LLMs to infer the structural information from the SMILES string, with the prompt described in Table 5.

Table 4: **Prompts for structure information analysis given SMILES string.**

---

**Head prompt:** You are now working as an excellent expert in chemistry and drug discovery. Given the SMILES representation of a molecule, your job is to predict the structural information of the molecule.
The structural information of the molecule caption includes the molecular formula, the length of the longest carbon chain, the number of aromatic rings, the IUPAC name of all the rings, all the functional groups, the number of chiral centers with S and R configurations each, the molecular weight, the IUPAC name of the molecule.
The functional group and ring IUPAC names should be on the list. The number of chiral centers should also be format {"S": , "R": }.
Your response should only be in the JSON format following {"molecular formula": , "functional group": , "longest carbon chain length": , "aromatic ring": , "ring IUPAC name":, "chiral": {"S": , "R": }, "weight": , "IUPAC name": }.
THERE SHOULD BE NO OTHER CONTENT INCLUDED IN YOUR RESPONSE. DO NOT CHANGE THE JSON KEY NAMES.

**Input prompt:** Input: <SMILES>

---

**Prompts given text description of molecules.** Next, we asked LLMs to infer the structural information from the text description of the molecule, with the prompt described in Table 4.

Table 5: **Prompts for structure information analysis given text description.**

---

**Head prompt:** You are now working as an excellent expert in chemistry and drug discovery. Given the caption of a molecule, your job is to predict the structural information of the molecule.
The molecule caption is a sentence that describes the molecule, which mainly describes the molecules´ structures, properties, and production.
The structural information of the molecule caption includes the molecular formula, the length of the longest carbon chain, the number of aromatic rings, the IUPAC name of all the rings, all the functional groups, the number of chiral centers with S and R configurations each, the molecular weight, the IUPAC name of the molecule.
The functional group and ring IUPAC names should be on the list. The number of chiral centers should also be format {"S": , "R": }.
Your response should only be in the JSON format following {"molecular formula": , "functional group": , "longest carbon chain length": , "aromatic ring": , "ring IUPAC name":, "chiral": {"S": , "R": }, "weight": , "IUPAC name": }.
THERE SHOULD BE NO OTHER CONTENT INCLUDED IN YOUR RESPONSE. DO NOT CHANGE THE JSON KEY NAMES.

**Input prompt:** Input: <Description>

---

## A.2 MOLECULE CAPTIONING

Here, we describe the detailed settings for the experiments of molecule captioning in Section 5.1. Note that we used four A100-80GB GPUs for fine-tuning.

**Hyperparameters.** The hyperparameters for the specialist models are provided in Table 6. Note that MolT5-large was not trained for the same number of epochs as the other models due to limited time constraints.

| Hyperparameter | MolT5-base | MolT5-large | ChemT5-small | ChemT5-base |
|---|---|---|---|---|
| Batch size | 8 | 4 | 8 | 8 |
| Learning rate | $2e^{-4}$ | $2e^{-4}$ | $6e^{-4}$ | $6e^{-4}$ |
| Epochs | 250 | 220 | 250 | 250 |
| Warmup ratio | 0 | 0 | 0.1 | 0.1 |
| Weight decay | 0.01 | 0.01 | 0 | 0 |
| Lr scheduler | linear | linear | linear | linear |

Table 6: **Hyperparameters for molecule captioning.**

**Prompts.** The prompts used for the generalist models are described in Table 10. We primarily followed the prompt presented by Li et al. (2024a).

---

**Head prompt:** You are now working as an excellent expert in chemistry and drug discovery. Given the SMILES representation of a molecule and structural description of the molecule, your job is to predict the caption of the molecule.
The molecule caption is a sentence that describes the molecule, which mainly describes the molecule's structures, properties, and production.

Example 1:
Instruction: Given the SMILES representation of a molecule, predict the caption of the molecule.
Input: <SMILES><STRUCTCOT >
Your output should be: {"caption": <Description>}
. . .
Example $k$:
Instruction: Given the SMILES representation of a molecule, predict the caption of the molecule.
Input: <SMILES><STRUCTCOT >
Your output should be: {"caption": <Description>}

Your response should only be in the JSON format above; THERE SHOULD BE NO OTHER CONTENT INCLUDED IN YOUR RESPONSE.

**Input prompt:** Input: <SMILES><STRUCTCOT >

---

Table 7: **Prompts for generalist models in text-based molecule generation task.**

## A.3 TEXT-BASED MOLECULE GENERATION

Here, we described the detailed settings for the experiments of text-based molecule generation in Section 4.3. Note that we also used four A100-80GB GPUs for fine-tuning.

**Hyperparameters.** The hyperparameters for the reasoning and answering module for the specialist models are provided in Table 8 and Table 9, respectively. Note that MolT5-large was not trained for the same number of epochs as the other models due to limited time constraints.

Table 8: **Hyperparameters for the reasoning module of text-based molecule generation.**

| Hyperparameter | MolT5-base | ChemT5-small | ChemT5-base |
|---|---|---|---|
| Batch size | 8 | 8 | 8 |
| Learning rate | $1e^{-3}$ | $6e^{-4}$ | $6e^{-4}$ |
| Epochs | 250 | 250 | 250 |
| Warmup ratio | 0.1 | 0 | 0 |
| Weight decay | 0 | 0 | 0 |
| Lr scheduler | cosine | linear | linear |

Table 9: **Hyperparameters for the answering module of text-based molecule generation.**

| Hyperparameter | MolT5-base | MolT5-large | ChemT5-small | ChemT5-base |
|---|---|---|---|---|
| Batch size | 8 | 4 | 8 | 8 |
| Learning rate | $1e^{-3}$ | $1e^{-3}$ | $6e^{-4}$ | $6e^{-4}$ |
| Epochs | 250 | 140 | 250 | 250 |
| Warmup ratio | 0.1 | 0.1 | 0 | 0 |
| Weight decay | 0 | 0 | 0 | 0 |
| Lr scheduler | cosine | cosine | linear | linear |

**Prompts.** The prompts used for the generalist models are described in Table 7. We also primarily followed the prompt presented by Li et al. (2024a).

Table 10: **Prompts for the generalist models in molecule captioning task.**

**Head prompt:** You are now working as an excellent expert in chemistry and drug discovery. Given the caption of a molecule, your job is to predict the SMILES representation of the molecule. The molecule caption is a sentence that describes the molecule, which mainly describes the molecule's structures, properties, and production.
You can infer the molecule SMILES representation from the caption.
Before you infer the molecule SMILES representation, YOU SHOULD FIRST GENERATE the molecular formula, the length of the longest carbon chain, the number of aromatic rings, the IUPAC name of all the rings, all the functional groups, the number of chiral centers with S and R configurations each, the molecular weight, the IUPAC name of the molecule.

Example 1: Instruction: Given the caption of a molecule, predict the SMILES representation of the molecule.
Input: <Description><STRUCTCOT >
Your output should be: {"molecule": <SMILES>}
. . .
Example $k$: Instruction: Given the caption of a molecule, predict the SMILES representation of the molecule.
Input: <Description><STRUCTCOT >
Your output should be: {"molecule": <SMILES>}

You should FIRST generate the structural information following the examples above, and then provide the JSON format of the molecule SMILES based on that.
NOTE THAT THE SMILES REPRESENTATION MUST BE IN THE JSON format above {"molecule": }. THERE SHOULD BE NO OTHER CONTENT INCLUDED IN YOUR JSON. DO NOT CHANGE THE JSON KEY NAME.

**Input prompt:** Input: <Description>

## A.4 ABLATION STUDY

Here, we describe the detailed settings for the ablation study.

**Prompts for ChemCrow.** The prompts used for ChemCrow (M. Bran et al., 2024) are described in Table 11 and Table 12. Notably, it was not able to apply few-shot learning for ChemCrow as it was not applicable as the original prompt proposed in ChemCrow does not include any few-shot setting.

Table 11: **Prompts for molecule captioning with ChemCrow.**

**Head prompt:** Given the SMILES representation of a molecule and structural description of the molecule, your job is to predict the caption of the molecule.
"Final Answer" follows the format: Final Answer: {"caption": }

**Input prompt:** The SMILES representation of the molecule is as follows: : <SMILES>

Table 12: **Prompts for text-based molecule generation with ChemCrow.**

**Head prompt:** Given the caption of a molecule, your job is to predict the SMILES representation of the molecule.
The molecule caption is a sentence that describes the molecule, which mainly describes the molecule's structures, properties, and production.
You can infer the molecule SMILES representation from the caption.
"Final Answer" follows the format: Final Answer: {"molecule": }

**Input prompt:** The caption is as follows: <Description>

## B  ADDITIONAL EXPERIMENTAL RESULTS

In this section, we provide additional experimental results including several concrete examples of generated samples.

### B.1  MOLECULE CAPTIONING

Here, we show the samples of molecule captioning, i.e., generated text descriptions of given molecules in Figure 9. Notably, we show the generated samples from base-sized models for fair comparison.

Figure 9: **The generated samples of molecule captioning.**

## B.2 TEXT-BASED MOLECULE GENERATION

Here, we show the samples of text-based molecule generation, i.e., generated molecules for the given text description in Figure 10. Notably, we show the generated samples from base-sized models for fair comparison.

Figure 10: **The generated samples of text-based molecule generation.**

Additionally, we provide the results of generalist models in Table 13. Note that it is natural to show no consistent enhancement for generalist models as they lack reasoning ability as shown in Table 2.

Table 13: **Text-based Molecule Generation Performance for generalist models.** The teal color indicates the improvement while the red color indicates the reduction.

| Models | BLEU | | Exact | | Levenshtein ↓ | | MACCS FTS | | RDK FTS | | Morgan FTS | | FCD↓ | | Validity | |
|---|---|---|---|---|---|---|---|---|---|---|---|---|---|---|---|---|
| | Met. | Δ | Met. | Δ | Met. | Δ | Met. | Δ | Met. | Δ | Met. | Δ | Met. | Δ | Met. | Δ |
| *Generalists (10-shot learning)* | | | | | | | | | | | | | | | | |
| Llama3 | 0.251 | - | 0.007 | - | 117.30 | - | 0.586 | - | 0.352 | - | 0.276 | - | 13.11 | - | 0.629 | - |
| +STRUCTCOT | 0.259 | 0.008 | 0.008 | 0.001 | 109.77 | 7.53 | 0.579 | 0.007 | 0.279 | 0.073 | 0.344 | 0.068 | 4.47 | 8.64 | 0.669 | 0.040 |
| GPT-4o | 0.521 | - | 0.079 | - | 40.87 | - | 0.797 | - | 0.496 | - | 0.583 | - | 3.67 | - | 0.881 | - |
| +STRUCTCOT | 0.509 | 0.012 | 0.088 | 0.009 | 41.68 | 0.081 | 0.783 | 0.014 | 0.498 | 0.002 | 0.571 | 0.012 | 1.57 | 2.10 | 0.846 | 0.035 |

## B.3 ABLATION STUDY

**Comparison to ChemCrow.** To validate the efficacy of our STRUCTCOT, we compare our method with ChemCrow (M. Bran et al., 2024), which has employed CoTs for various chemical tasks. The comparative results are provided in Table 14 and Table 16. One can observe that ChemCrow shows limited performance in both molecule captioning and text-based molecule generation tasks. It is notable that the comparison may not be entirely appropriate, as ChemCrow is primarily designed for practical synthesis tasks, as the reviewer mentioned. Nevertheless, we included comparisons with ChemCrow to provide additional insights, as they share a similar motivation: enriching large language models (LLMs) with a chemistry-aware chain-of-thoughts.

Table 14: **Comparison to ChemCrow in molecule captioning.** The specialist model indicates our results from MolT5-large while the generalist model indicates the one from GPT-4o.

| Models | BLEU-2 | BLEU-4 | ROUGE-1 | ROUGE-2 | ROUGE-L | METEOR |
|---|---|---|---|---|---|---|
| ChemCrow (GPT-4o) | 0.162 | 0.078 | 0.299 | 0.097 | 0.211 | 0.225 |
| Ours (GPT-4o) | 0.249 | 0.139 | 0.386 | 0.179 | 0.300 | 0.303 |
| Ours (ChemT5-base) | 0.639 | 0.560 | 0.687 | 0.553 | 0.626 | 0.657 |

Table 15: **Comparison to ChemCrow in text-based molecule generation.** The specialist model indicates our results from GPT-4o while the generalist model indicates the one from .

| Models | BLEU | Exact | Levenshtein ↓ | MACCS FTS | RDK FTS | Morgan FTS | FCD ↓ | Validity |
|---|---|---|---|---|---|---|---|---|
| ChemCrow (GPT-4o) | 0.306 | 0.194 | 56.46 | 0.772 | 0.632 | 0.555 | 2.31 | 0.851 |
| Ours (GPT-4o) | 0.509 | 0.088 | 41.68 | 0.783 | 0.498 | 0.571 | 1.57 | 0.846 |
| Ours (ChemT5-base) | 0.878 | 0.421 | 12.76 | 0.924 | 0.856 | 0.804 | 0.26 | 0.982 |

**Structural component.** To verify the effectiveness of each component, we evaluated the performance of molecule captioning on ChemT5-small using each structural information component individually. We provide the results in Figure 11. We observe that the molecular formula and chirality contribute the largest performance improvements. Additionally, incorporating each single component resulted in better performance compared to the baseline model without any Chain-of-Thought (CoT) integration. Notably, combining all the proposed structural elements yielded the best results, validating the effectiveness of our comprehensive approach.

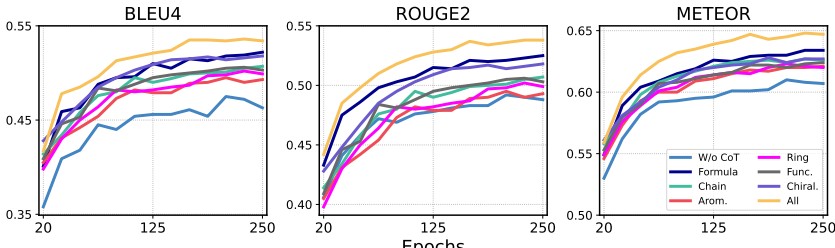

Figure 11: **The impact of each structural component.**

**Additional molecular descriptors.** In addition to the proposed six structural components, we conducted experiments using three more advanced molecular descriptors: the Morgan fingerprint and two electronic properties—topological polar surface area (TPSA) and molar refractivity (MR). Specifically, the Morgan fingerprint encodes local substructures within a specified radius; TPSA represents the sum of the surface areas of all polar atoms and their attached hydrogen atoms; and MR quantifies the total polarizability of a molecule.

To verify the effectiveness of each additional descriptor, we evaluated the performance of molecule captioning using ChemT5-small. We provide the results in Figure 12. We observed that incorporating all three additional descriptors together did not further improve the performance of StructCoT, although applying each additional descriptor individually improved performance. This validates the importance of structural information and the sufficiency of our proposed structural components.

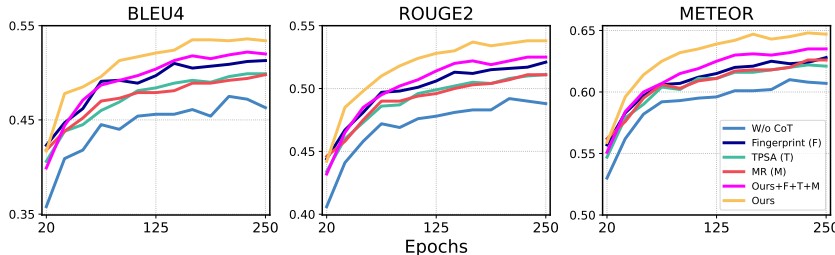

Figure 12: **The impact of additional molecular descriptors.**

## B.4 RETROSYNTHESIS

Here, we present additional experimental results on retrosynthesis with Mol-Instructions (Fang et al., 2024). We incorporate STRUCTCOT by prompting the model with the target molecule and STRUCTCOT. We observe that incorporating our STRUCTCOT not only improved the molecular captioning and text-based molecule generation tasks but also more complicated retrosynthesis task.

Table 16: **Comparison to Mol-Instructions in retrosynthesis.**

| Models | BLEU | Exact | Levenshtein ↓ | MACCS FTS | RDK FTS | Morgan FTS ↓ | Validity |
|---|---|---|---|---|---|---|---|
| Mol-Instructions | 0.009 | 0.705 | 31.23 | 0.283 | 0.487 | 0.230 | 1.000 |
| + STRUCTCOT | 0.016 | 0.502 | 31.21 | 0.315 | 0.493 | 0.273 | 1.000 |

