# OpenReview forum: "Chain-of-thoughts for molecular understanding"
_ICLR.cc/2025/Conference — Submitted to ICLR 2025_

### Official Review · Reviewer_GaNM · 2024-10-27

**Soundness:** 2
**Presentation:** 3
**Contribution:** 3
**Rating:** 6
**Confidence:** 3

**Summary:**

Summary:

The paper proposed StructCoT - a structure-aware chain-of-thought method including six predefined structure information elements as part of the input to LLMs. The paper also proposed a matching ratio-based rejection sampling method to help sample structure valid molecules.

**Strengths:**

Strengths:

1. Introducing structure information in LLM-based molecule understanding is crucial.
2. The paper is well-written and easy to understand.

**Weaknesses:**

Weaknesses:

1. The matching ratio-based rejection sampling requires further clarification and expansion. For instance, it is unclear whether different structural elements are weighted differently when calculating the matching ratio. In the experiments, the authors exclude molecular formula, molecular weight, and IUPAC names due to their low reasoning accuracies. It is also necessary to explore how the parameter k influences the sampling performance; while k=5 is used in the ablation study, the impact of varying k remains unexamined. Furthermore, the influence of the sampling module on efficiency should be addressed. Discussing the efficiency and running time associated with this sampling method would be beneficial for understanding its practical implications.
2. Further ablation studies about how different structural information impacts performance are necessary to conduct. As a main contribution of the paper, six structural elements are incorporated within the methodology. However, the paper currently lacks a detailed analysis of how each individual element influences performance. It is beneficial to include these studies in the experimental section.

**Questions:**

Please refer to weaknesses

---

> ### Author Response · Authors · 2024-11-22
>
> We sincerely appreciate your comments and efforts in reviewing our paper. We think our paper has improved significantly by adding a comprehensive ablation study. We address your question as follows. We also updated our manuscript by highlighting the updates in $\color{magenta}{\text{magenta}}$.
>
> ---
>
> **W1. The matching ratio-based rejection sampling needs further clarification including the weights between structural components, the impact of the number of samples $k$, and the efficiency or the running time of the rejection sampling.**
>
> Thank you for pointing out the unclarity. First, the weights between structural components are not adjusted and all components are treated equally.
>
> Next, for the impact of the number of samples $k$, we conducted an additional ablation study on $k\in\{10,20\}$. It is notable that the results for $k=1$ (i.e., no matching ratio-based sampling) were already presented in Figure 6. We updated Figure 8 in our manuscript to include the additional results for $k\in\{10,20\}$. We observe that while applying matching ratio-based rejection sampling consistently improves performance, increasing $k$ beyond 5 does not result in significant further improvement. Therefore, our original choice of $k=5$ was sufficient to balance performance and efficiency.
>
> Lastly, regarding the efficiency of the rejection sampling, the rejection sampling does not impact training efficiency, as it only influences the sampling process. During generation, the approach leverages normal beam search to generate multiple outputs. The time complexity of this process remains the same as the original beam search, which is O(d \times b \times \log(b)), where d is the number of successors per node, and b is the number of levels to explore. This ensures that the method is computationally efficient and scalable.
>
> ---
> **W2. An ablation study on each structural information component is needed.**
>
> Thank you for such an insightful comment! Following your suggestion, we conducted an ablation study on each structural information component. We report the new results in Figure 11 of Appendix B.3.
>
> In this updated analysis, we evaluated the performance of molecule captioning on ChemT5-small using each structural information component individually. The results show that the molecular formula and chirality contribute the largest performance improvements. Additionally, incorporating each single component resulted in better performance compared to the baseline model without any Chain-of-Thought (CoT) integration. Notably, combining all the proposed structural elements yielded the best results, validating the effectiveness of our comprehensive approach.

---

> > ### Comment · Reviewer_GaNM · 2024-11-26
> > **Response to rebuttal**
> >
> > I would like to thank the authors for their detailed explanations and efforts. I will maintain my positive score and am inclined to accept the paper.

---

> > > ### Author Response · Authors · 2024-12-02
> > >
> > > Thank you for your insightful comments, which have significantly enhanced our paper. We sincerely appreciate your valuable feedback!

---

### Official Review · Reviewer_atZJ · 2024-10-31

**Soundness:** 2
**Presentation:** 2
**Contribution:** 2
**Rating:** 3
**Confidence:** 5

**Summary:**

The authors propose StructCoT, a structure-aware chain-of-thought (CoT), to improve the LLMs' understanding of molecular structures by directly pointing out the crucial features in their structures. At the same time, two fine-tuing frameworks are also incorporated for StructCoT. Experimental results show that STructCoT can improve the performance in molecular understanding tasks such as molecule captioning.
Soundness:

**Strengths:**

1. Structural information, epecially the substurture information, is important for accurate molecule understanding.
2. StructCoT does contribute to the performance improvement according to the experimental results.

**Weaknesses:**

1. The novelty is not enough. Chain-of-thought (CoT) is widely used in prompting LLMs for complex tasks. I don't see enough technical contributions in the proposed method.
2. The presentation of this paper is poor. Firstly, the authors claim that StructCoT could improve the molecule understanding tasks, but in the experiments, they include the text-based molecule generation task, which does not belong to the molecule understanding task. Meanwhile, how could you obtain accurate molecule structural information if LLMs are not reliable as you claim previously in Figure 1? Secondly, after reading through this paper, I still could not figure out clearly what "two fine-tuning frameworks" mean, especailly in Figure 2. Does the authors mean two tasks? Thirdly, in the molecule captioning task, It seems that no CoT is applied, and the structral information is generated by external tools like Rdkit. Forthly, the xlabel of Figure 6 is missing.
3. The performance of GPT-4o in molecule captioning task is way too low. According to MolReGPT, the 10-shot performance of GPT-4 is comparable to MolT5-large. As GPT-4o is even more advanced, I remain doubtful on the experiment results. Did the authors follow the prompt template used in MolReGPT?
4. Although BioT5 is much better in their original performance, the authors still choose the ChemT5. I wonder what is the exact reason.
5. The baselines selected are not strong enough, please include more up-to-date methods for comparison.

**Questions:**

Please address the weaknesses mentioned above.

---

> ### Author Response · Authors · 2024-11-22
>
> We sincerely appreciate your comments and efforts in reviewing our paper. We think our paper has improved significantly in terms of additional baseline, thanks to your comment. We address your question as follows. We also updated our manuscript by highlighting the updates in $\color{magenta}{\text{magenta}}$.
>
> ---
> **W1. The method is not novel as CoT is widely used in prompting LLMs for complex tasks.**
>
> We respectfully disagree with the assessment that it lacks novelty. We have addressed a key bottleneck in chemistry tasks. Even recent strong LLMs have shown limitations in chemistry tasks. Our work qualitatively identifies this bottleneck as the lack of capability to understand molecular structures and proposes StructCoT to integrate molecular structural information that directly relates to significant molecular properties. This innovation addresses a critical gap in applying LLMs to chemistry.
>
> While we acknowledge that Chain-of-Thought (CoT) reasoning has been widely adopted in prompting LLMs for various complex tasks, our contributions specifically address unique challenges in the domain of molecular reasoning, which remain largely underexplored. Unlike natural language tasks, molecular representations (e.g., SMILES) lack inherent structural context, which directly impacts the properties of the molecules. StructCoT not only incorporates this missing structural information but also demonstrates its importance for enhancing the reasoning capabilities of LLMs in molecular tasks.
>
> Moreover, a distinct contribution of StructCoT is its ability to assess the alignment between the generated structural rationale and the corresponding molecule. This enables a matching ratio-based rejection sampling mechanism, a novel approach to ensuring consistency of the rationale and the output. To the best of our knowledge, this is the first work to propose evaluating the coherence between a generated rationale and the final answer in molecular reasoning tasks.
>
> ---
> **W2-1. The meaning of molecular understanding is unclear as text-based molecule generation is not an understanding task.**
>
> We used the term "molecular understanding" as both molecular captioning and text-based molecule generation tasks require a deep understanding of the molecules. We clarified this in the first paragraph of the introduction that molecular understanding tasks in our context refer specifically to both molecule captioning and text-based molecule generation.
>
> We used the term “molecular understanding” because both molecular captioning and text-based molecule generation tasks require a deep understanding of molecular structures and properties. These tasks necessitate a comprehensive understanding of the underlying molecular information.
>
>
> **W2-2. The meaning of "two fine-tuning frameworks" is unclear.**
>
> As mentioned in Figure 2, Sections 4.2 and 4.3, this means that we use two different frameworks for molecule captioning and text-based molecule generation.
>
> **W2-3. The word CoT is improper for molecular captioning task as it is generated by external tools.**
>
> First, we clarify that StructCoT is generated by the LLM during the text-based molecule generation task, which aligns with the general meaning of CoT as intermediate reasoning steps generated by the model.
>
> In addition, our use of the term "CoT" follows prior works [3,5,6], where the CoT process has been extended to include reasoning enhanced by external tools. For instance,
> - ChemCrow [3] employed external tools such as literature search and SMILES to price to generate proper reasoning for LLM.
> - MultiTool-CoT [5] proposed to incorporate multiple external tools such as a calculator and knowledge retrieval during the reasoning process.
> - PoT [6] introduced to incorporate a program interpreter into the CoT generation process.
>
> These works demonstrate that incorporating external tools into the reasoning process does not conflict with the CoT framework. However, if you have alternative suggestions for terminology that better aligns with the context, we are open to adopting them.
>
> **W2-4. The x-axis label of Figure 6 is missing.**
>
> Thank you for pointing this out. The x-axis label represents epochs, consistent with Figure 8. We have added the x-axis label in the updated manuscript to ensure clarity.

---

> ### Author Response · Authors · 2024-11-22
>
> **W3. The molecular captioning performance of GPT-4o is too low compared to that of MolReGPT. Did the authors follow the prompt template used in MolReGPT?**
>
> We confirm that we followed the prompt template used in MolReGPT, as detailed in Table 8 and Table 10 of Appendix A.2 and A.3. It is notable that while we mostly adhered to the original prompt, we made slight modifications to incorporate the StructCoT. This adjustment ensures that the structural reasoning process is effectively introduced and leveraged within our framework. Despite adhering to MolReGPT’s prompt design, the reported molecular captioning performance of GPT-4o is lower than that of MolReGPT. This discrepancy arises because we did not apply any retrieval augmentation approach, which is a key component of MolReGPT’s methodology.
>
> It is important to note that retrieval augmentation is orthogonal to our approach. Incorporating our StructCoT framework into MolReGPT could be an interesting direction for future work, potentially combining the strengths of both CoT and retrieval augmentation to further improve the molecular captioning performance.
>
> ---
> **W4. The baseline BioT5 is missing.**
>
> We regret that we are unable to conduct the experiments within the rebuttal phase due to computational limitations. In detail, fine-tuning BioT5 requires 100,000 steps with a batch size of 768, which could not be completed within the limited time and resources as it requires several hundred epochs. We plan to incorporate these experiments into our future manuscript. Although we are not able to incorporate StructCoT into BioT5, it is notable that we have already achieved competitive results with BioT5 in both tasks by incorporating ChemT5-base with our proposed StructCoT.
>
> ---
> **W5. The selected baselines are outdated and not strong enough.**
>
> To address your concern, we have conducted additional experiments using the recent MolInstructions [7] dataset for molecule captioning. The results detailed below and in Table 1, demonstrate that our approach also achieves improved performance on MolInstructions.
>
> |                  | BLEU-2 | BLEU-4 | ROUGE-1 | ROUGE-2 | ROUGE-L | METEOR |
> | ---------------- | ------ | ------ | ------- | ------- | ------- | ------:|
> | Mol-Instructions | 0.217  | 0.143  | 0.337   | 0.196   | 0.291   |  0.254 |
> | +StructCoT       | 0.347  | 0.275  | 0.601   | 0.518   | 0.593   |  0.520 |
>
>
> Moreover, we have also conducted additional experiments on retrosynthesis to verify the effectiveness of our method for other tasks. The results are detailed below and in Appendix B.4. We observe that incorporating our StructCoT not only improves molecular captioning and text-based molecule generation tasks but also more complicated retrosynthesis task.
>
>
> |                  | Exact | BLEU  | Leven. (↓) | RDK FTS | MACCS FTS       | Morgan FTS | Val.    |
> | ---------------- | ----- | ----- | -------------- | ------- | --------------- | ---------- | --- |
> | Mol-Instructions | 0.009 | 0.705 | 31.23          | 0.283   | 0.487           |        0.230    | 1.000    |
> | +StructCoT       | 0.016 | 0.502 | 31.21          | 0.315   | 0.493    |  0.273          | 1.000    |
>
>
> ---
> [1] Bran, A. M., et al. "ChemCrow: Augmenting large-language models with chemistry tools." Nature Machine Intelligence 2024.
>
> [2] Devlin, J., et al. "Bert: Pre-training of deep bidirectional transformers for language understanding." Proceedings of NAACL-HLT 2019.
>
> [3] Bran, A. M., et al. "ChemCrow: Augmenting large-language models with chemistry tools." Nature Machine Intelligence 2024.
>
> [4] Radford, A., et al. "Improving Language Understanding by Generative Pre-Training (GPT-1)." Arxiv 2018.
>
> [5] Inaba, T., et al. "MultiTool-CoT: GPT-3 can use multiple external tools with chain of thought prompting." ACL 2023.
>
> [6] Xenhu Chen, et al., "Program of Thoughts Prompting: Disentangling Computation from Reasoning for Numerical Reasoning Tasks." TMLR 2023.
>
> [7] Fang, Yin, et al. "Mol-instructions: A large-scale biomolecular instruction dataset for large language models." ICLR 2024.

---

> > ### Author Response · Authors · 2024-12-02
> >
> > Dear reviewer atZJ,
> >
> > Thank you for your time and efforts in reviewing this paper. Since the discussion phase is close to the end, we would like to inquire if our responses have addressed your concerns.
> >
> > We are especially curious since you indicated that your assessment of our paper was negative and we would like to change the score if your concerns are addressed. We wonder if our response has successfully addressed the issues.
> >
> > We remain fully committed to addressing any questions you may have by the end of the discussion phase.

---

### Official Review · Reviewer_gWgH · 2024-11-03

**Soundness:** 2
**Presentation:** 2
**Contribution:** 1
**Rating:** 5
**Confidence:** 5

**Summary:**

The paper addresses the limitations of large language models (LLMs) like GPT-4 in accurately interpreting and reasoning about molecular structures. The authors introduce StructCoT, a structure-aware chain-of-thought framework designed to enhance LLMs' performance on molecular understanding tasks. StructCoT explicitly incorporates key molecular structural features into the reasoning process, aiming to improve tasks such as molecule captioning (Mol2Text) and text-based molecule generation (Text2Mol). The authors also propose fine-tuning frameworks to adapt existing LLMs to effectively utilize StructCoT, demonstrating consistent performance improvements over baseline models.

**Strengths:**

The paper tackles the novel challenge of integrating detailed molecular structural information into the reasoning process of LLMs. The paper is well-organized and clearly articulates the motivation behind StructCoT. The methodology is systematically presented, with clear definitions of the six key structural elements incorporated into StructCoT. The experimental setup is thorough, and the results show that incorporating StructCoT leads to performance improvements in the evaluated tasks.

**Weaknesses:**

- Limited Evaluation Scope: The paper evaluates StructCoT primarily on tasks involving the generation and interpretation of molecular descriptions (Mol2Text and Text2Mol). These tasks, while relevant, may not fully capture the complexity of molecular understanding. Evaluating the model on more challenging tasks, such as molecular property prediction, reaction prediction, or synthesis planning, would provide a more comprehensive assessment of its capabilities.

- Methodology too Simple: The methodology relies heavily on augmenting input data with explicit structural information extracted using existing tools like RDKit. While this approach is practical, it raises questions about the novelty of the contribution. The performance improvements might stem more from the inclusion of precise structural data rather than the proposed StructCoT framework itself.

- Underutilization of Advanced Tools: Given that RDKit and similar tools can compute complex molecular properties and descriptors, the paper's focus on basic structural features seems limiting. Incorporating more advanced concepts, such as molecular fingerprints, electronic properties, or 3D conformations, could enhance the model's understanding and enable it to tackle more sophisticated tasks.

**Questions:**

1. Have the authors considered testing StructCoT on more complex molecular understanding tasks, such as molecular property prediction or reaction outcome prediction? How does StructCoT perform in these contexts compared to existing methods?

2. Since RDKit can compute a wide range of molecular descriptors and properties, why did the authors limit the structural information to the six basic elements? Would incorporating more complex concepts, as suggested in AutoMolCo [1], lead to better performance or broader applicability?

[1] Zhang, Shichang, et al. "Automated Molecular Concept Generation and Labeling with Large Language Models." arXiv preprint arXiv:2406.09612 (2024).

---

> ### Author Response · Authors · 2024-11-22
>
> We sincerely appreciate your comments and efforts in reviewing our paper. We think our paper has improved significantly in terms of additional tasks and advanced structural components, thanks to your comment. We address your question as follows. We also updated our manuscript by highlighting the updates in $\color{magenta}{\text{magenta}}$.
>
> ---
> **W1, Q1. Incorporating various tasks such as property prediction, reaction prediction, and synthesis planning may strengthen the paper.**
>
> Thank you for your insightful suggestions. To address your comments, we have conducted additional experiments on retrosynthesis to verify the effectiveness of our method for other tasks. The results are detailed below and in Appendix B.4. We observe that incorporating our StructCoT not only improves molecular captioning and text-based molecule generation tasks but also more complicated retrosynthesis task.
>
>
> |                  | Exact | BLEU  | Leven. (↓) | RDK FTS | MACCS FTS       | Morgan FTS | Val.    |
> | ---------------- | ----- | ----- | -------------- | ------- | --------------- | ---------- | --- |
> | Mol-Instructions | 0.009 | 0.705 | 31.23          | 0.283   | 0.487           |        0.230    | 1.000    |
> | +StructCoT       | 0.016 | 0.502 | 31.21          | 0.315   | 0.493    |  0.273          | 1.000    |
>
>
> ---
> **W2. While the proposed approach is practical, the method itself is too simple.**
>
> We acknowledge that our approach is straightforward; however, we respectfully disagree with the assessment that simplicity is our weakness. We have addressed a key bottleneck in chemistry tasks. Even recent strong LLMs have shown limitations in chemistry tasks. Our work qualitatively identifies this bottleneck as the lack of capability to understand molecular structures and proposes StructCoT to integrate molecular structural information that directly relates to significant molecular properties. This innovation addresses a critical gap in applying LLMs to chemistry.
>
> ---
>
> **W3, Q2. What is the reason for choosing the six proposed structural information components? Incorporating more advanced concepts (e.g., molecular fingerprints, electronic properties, 3D conformations, etc.) may improve the method.**
>
> Thank you for your insightful suggestions. In our work, we selected six "structural" features of molecules as they are highly relevant to molecular properties, as illustrated in Figure 3. Nevertheless, to address your comments, we conducted additional experiments incorporating three advanced components: fingerprint information (Morgan fingerprint) and electronic properties (topological polar surface area (TPSA) and molar refractivity (MR)). Specifically, the Morgan fingerprint encodes local substructures within a specified radius, TPSA represents the sum of the surface areas of all polar atoms and their attached hydrogen atoms, and MR quantifies the total polarizability of a molecule.
>
> To verify the effectiveness of each additional descriptor, we evaluated the performance of molecule captioning using ChemT5-small. We provide the results in Figure 12 of Appendix B.3. We observed that incorporating all three additional descriptors together did not further improve the performance of StructCoT, although applying each additional descriptor individually improved performance. This validates the importance of structural information and the sufficiency of our proposed structural components.

---

> > ### Author Response · Authors · 2024-12-02
> >
> > Dear reviewer gWgH,
> >
> > Thank you for your time and efforts in reviewing this paper. Since the discussion phase is close to the end, we would like to inquire if our responses have addressed your concerns.
> >
> > We are especially curious since you indicated that your assessment of our paper was borderline and we would like to change the score if your concerns are addressed. We wonder if our response has successfully addressed the issues.
> >
> > We remain fully committed to addressing any questions you may have by the end of the discussion phase.

---

### Official Review · Reviewer_ZbGe · 2024-11-03

**Soundness:** 3
**Presentation:** 3
**Contribution:** 3
**Rating:** 6
**Confidence:** 3

**Summary:**

This study introduces STRUCTCOT, a structure-aware chain-of-thought (CoT) approach designed to enhance large language models (LLMs) for molecular structure understanding tasks. By embedding key structural features into LLMs, STRUCTCOT aims to improve the models' performance in chemistry-specific applications. The results show that incorporating STRUCTCOT consistently enhances molecular understanding tasks, highlighting its potential to address a notable gap in LLM-based molecular reasoning.

**Strengths:**

The study presents an innovative approach by focusing on structural feature embedding, a novel strategy for leveraging LLMs in chemistry. The method demonstrates notable improvements in molecular understanding tasks, suggesting that STRUCTCOT could effectively bridge limitations of traditional LLMs in molecular reasoning. The work’s originality and the clear effort involved in developing STRUCTCOT make it a valuable contribution to the field of ai for chemistry.

**Weaknesses:**

One potential limitation is the choice to use an LLM-based approach for molecular structure tasks, where specialized models like Chemprop[1] are typically more efficient and effective. Additionally, existing tools like RDKit can generate much of the necessary structural information directly, often without the interpretative challenges LLMs face with SMILES strings. These aspects raise questions about whether LLMs are the most suitable tool for these tasks, considering their current limitations.

[1] Heid, Esther, et al. "Chemprop: a machine learning package for chemical property prediction." Journal of Chemical Information and Modeling 64.1 (2023): 9-17.

**Questions:**

What is the primary motivation for adapting LLMs to structure-based molecular tasks, given the availability of specialized models (e.g., Chemprop) and tools (e.g., RDKit) that handle structural information efficiently?
How were the specific structural features selected for inclusion in STRUCTCOT? For example, complex configurations like chirality and E/Z isomerism might be critical in some molecular contexts, but it’s unclear whether and how these were considered in the study’s framework.

---

> ### Author Response · Authors · 2024-11-22
>
> We sincerely appreciate your comments and efforts in reviewing our paper. We think our paper has improved significantly in terms of advanced structural components, thanks to your comment. We address your question as follows. We also updated our manuscript by highlighting the updates in $\color{magenta}{\text{magenta}}$.
>
> ---
>
> **W1, Q1-1. Why do we need to employ general LLMs for molecular understanding tasks? Specialist models such as Chemprop may work better and external tools are more appropriate for acquiring structural information.**
>
> General large language models (LLMs), such as Llama and GPT-4o, excel in their ability to generalize across diverse domains, enabling them to adapt to a wide range of tasks that specialist models may not address effectively. Molecular understanding tasks, for instance, often require interpreting textual descriptions of molecules or integrating molecular reasoning with external knowledge. Generalist models, pre-trained on diverse text corpora, including scientific literature, are well-suited for these challenges. While the current performance of generalist models in some molecular tasks may lag behind specialist models, we anticipate that advancements in computational power and modeling capabilities will enable LLMs to surpass specialist models in the near future.
>
>
> Moreover, we clarify that the integration of external tools to enrich structural information complements rather than replaces the utility of generalist models. StructCoT bridges this gap by enhancing generalist models with molecular structural information, leveraging their reasoning capabilities while addressing their limitations in understanding structural features. This integrated approach enables a broader application of LLMs in molecular understanding tasks, without being restricted by the narrower focus of specialist models.
>
>
> ---
>
> **Q1-2. What is the reason for choosing the six proposed structural information components? Incorporating more advanced concepts such as chirality and E/Z isomerism may improve the method.**
>
> Thank you for your insightful suggestions. In our work, we selected six "structural" features of molecules as they are highly relevant to molecular properties, as illustrated in Figure 3. Nevertheless, to address your comments, we conducted additional experiments incorporating three advanced components: fingerprint information (Morgan fingerprint) and electronic properties (topological polar surface area (TPSA) and molar refractivity (MR)). Specifically, the Morgan fingerprint encodes local substructures within a specified radius, TPSA represents the sum of the surface areas of all polar atoms and their attached hydrogen atoms, and MR quantifies the total polarizability of a molecule.
>
> To verify the effectiveness of each additional descriptor, we evaluated the performance of molecule captioning using ChemT5-small. We provide the results in Figure 12 of Appendix B.3. We observed that incorporating all three additional descriptors together did not further improve the performance of StructCoT, although applying each additional descriptor individually improved performance. This validates the importance of structural information and the sufficiency of our proposed structural components.
>
> ---
>
> [1] Edwards, C., et al. "Translation between molecules and natural language." ACL 2022.
>
> [2] Christofidellis, D., et al. "Unifying molecular and textual representations via multi-task language modeling." ICML 2023.

---

> > ### Author Response · Authors · 2024-12-02
> >
> > Dear reviewer ZbGe,
> >
> > Thank you for your time and efforts in reviewing this paper. Since the discussion phase is close to the end, we would like to inquire if our responses have addressed your concerns.
> >
> > We remain fully committed to addressing any questions you may have by the end of the discussion phase.

---

### Official Review · Reviewer_NySt · 2024-11-03

**Soundness:** 2
**Presentation:** 2
**Contribution:** 2
**Rating:** 6
**Confidence:** 5

**Summary:**

The paper proposes a framework to enhance molecule captioning and generation by integrating structure-related textual information. In the molecule-to-text task, structural information is extracted using RDKit and then provided along with the SMILES representation as input to fine-tune models for caption generation. For the text-to-molecule task, an additional model is trained to generate this structural information from captions, which is subsequently passed to a target model to generate molecules. This approach demonstrates improved performance in molecule captioning and generation with models such as MolT5 and ChemT5.

**Strengths:**

- The proposed method of adding structural information is well-motivated and systematically applied, with distinct steps for molecule-to-text and text-to-molecule tasks. The overall setup is clearly presented.
- The method demonstrates encouraging results, particularly in the text-to-molecule task, where structural prompts improve fidelity to the exact SMILES representation. This suggests that the structural prompts positively impact molecule generation accuracy.
- The paper is generally clearly written and well-organized.

**Weaknesses:**

1. **Limited Dataset and Baselines**: The evaluation could benefit from using a larger and more diverse training set, such as PubChemSTM[1], which would add robustness to the results. Additionally, several recent models (such as the ones in MolInstructions[2]) are not benchmarked, limiting the scope of the evaluation.
2. **Comparisons with ChemCrow and BioT5**: The comparison with ChemCrow is not entirely appropriate, as ChemCrow was not specifically designed for molecule captioning. Additionally, the method underperforms compared to BioT5 in certain metrics. An augmented BioT5 would serve as a stronger baseline, potentially offering more relevant comparisons.
3. **Handling of Missing Structural Information in Text2Mol**: There is no mechanism for handling cases where structural details may be absent in the caption during text-to-molecule generation. This could lead to "hallucination" of inaccurate details, affecting the reliability of generated molecules.
4. **Limited Technical Contribution**: The technical novelty of the paper is modest, as the approach primarily focuses on input augmentation rather than introducing a new model architecture or algorithm.
5. **Narrow Scope of Structural Information**: The structural information used is restricted to basic elements like functional groups and ring systems. Given the diversity of molecular captions, expanding beyond these elements could increase the effectiveness of the method.
- Minor: The term “Chain of Thought” (CoT) is misleading in this context, as it traditionally refers to intermediate reasoning generated by the model. Here, “StructCoT” is provided as input rather than produced by the LLM. This choice of terminology may mislead readers, as the structural information is not generated by the model itself.
- Minor:  Phrasing like "We propose to" in the contributions section could imply a suggestion rather than a concrete result. It might be more effective to use definitive statements, such as “We demonstrate…”.

[1] Li, Sihang, et al. "Towards 3d molecule-text interpretation in language models." *arXiv preprint arXiv:2401.13923* (2024).

[2] Fang, Yin, et al. "Mol-instructions: A large-scale biomolecular instruction dataset for large language models." *arXiv preprint arXiv:2306.08018* (2023).

**Questions:**

- While the method positively impacts captions with structural information, it is not clear whether it aids captions that focus on other descriptors (e.g., molecular activity or properties). Are there any results on how the structural prompt affects captions beyond structural descriptions?
- BioT5 is noted to outperform the proposed approach in some tasks. Would incorporating the same structural augmentation improve BioT5’s performance further, potentially establishing a new baseline?
- Although the method improves molecule captioning performance, the real-world applications of these gains are unclear. Are there specific downstream tasks that would benefit directly from enhanced captioning? This is a question regarding the general line of research, and not specifically pointed at this paper.

---

> ### Author Response · Authors · 2024-11-22
>
> We sincerely appreciate your comments and efforts in reviewing our paper. We think our paper has improved significantly in terms of additional baseline and advanced structural component, thanks to your comment. We address your question as follows. We also updated our manuscript by highlighting the updates in $\color{magenta}{\text{magenta}}$.
>
> ---
>
> **W1. Incorporating additional baselines [2] may strengthen the paper.**
>
> Thank you for your insightful suggestions. To address your comments, we have conducted additional experiments using the MolInstructions dataset for molecule captioning. The results detailed below and in Table 1, demonstrate that our approach also achieves improved performance on MolInstructions.
>
> |                  | BLEU-2 | BLEU-4 | ROUGE-1 | ROUGE-2 | ROUGE-L | METEOR |
> | ---------------- | ------ | ------ | ------- | ------- | ------- | ------:|
> | Mol-Instructions | 0.217  | 0.143  | 0.337   | 0.196   | 0.291   |  0.254 |
> | +StructCoT       | 0.347  | 0.275  | 0.601   | 0.518   | 0.593   |  0.520 |
>
>
> Moreover, we have also conducted additional experiments on retrosynthesis to verify the effectiveness of our method for other tasks. The results are detailed below and in Appendix B.4. We observe that incorporating our StructCoT not only improves molecular captioning and text-based molecule generation tasks but also more complicated retrosynthesis task.
>
>
> |                  | Exact | BLEU  | Leven. (↓) | RDK FTS | MACCS FTS       | Morgan FTS | Val.    |
> | ---------------- | ----- | ----- | -------------- | ------- | --------------- | ---------- | --- |
> | Mol-Instructions | 0.009 | 0.705 | 31.23          | 0.283   | 0.487           |        0.230    | 1.000    |
> | +StructCoT       | 0.016 | 0.502 | 31.21          | 0.315   | 0.493    |  0.273          | 1.000    |
>
> ---
>
> **W2, Q2. Comparison to BioT5 [4] could be additional baseline and Chemcrow [3] is not entirely appropriate as it is designed for different goals.**
>
> Regarding BioT5, we regret that we are unable to conduct the experiments within the rebuttal phase due to computational limitations. In detail, fine-tuning BioT5 requires 100,000 steps with a batch size of 768, which could not be completed within the limited time and resources as it requires several hundred epochs. We plan to incorporate these experiments into our future manuscript. Although we are not able to incorporate StructCoT into BioT5, it is notable that we have already achieved competitive results with BioT5 in both tasks by incorporating ChemT5-base with our proposed StructCoT.
>
> For ChemCrow, we acknowledge that the comparison may not be entirely appropriate, as the reviewer mentioned, ChemCrow is primarily designed for practical synthesis tasks, but we included the comparisons to provide additional insights, as they share a similar motivation. To alleviate your concern, we added a discussion on the appropriateness of comparing to ChemCrow in Appendix B.3.
>
>
> ---
>
> **W3, Q1. There is no mechanism for handling missing structural information in captions during text-based molecule generation. How does the structural CoT affect captions beyond structural information?**
>
> In fact, StructCoT is specifically designed to address such scenarios. As detailed in Figure 2(b) and Section 4.3, we train the reasoning module to infer structural information from the given caption, regardless of whether explicit structural details are present in the caption. This approach ensures robustness in cases where structural information is incomplete or absent in the input caption.
>
> Moreover, sketching the structural description before output generation enhances the model's understanding of molecular structures, effectively bridging gaps in the input. This improved structural awareness positively impacts the overall quality of the generated outputs, even in cases where the captions are less relevant to the structural description. As a result, by prioritizing structural reasoning, StructCoT enables more accurate outputs, demonstrating its effectiveness.

---

> ### Author Response · Authors · 2024-11-22
>
> **W4. The technical novelty of the method is modest as the approach focuses on input augmentation rather than proposing a new model architecture.**
>
> We acknowledge the reviewer's observation but would like to clarify the scope and contributions of our work. While technical novelty is important in research, enhancing practical performance through straightforward approaches, such as input augmentation and constructing new datasets, is also a critical area of innovation. For instance, developing new datasets and benchmarks is widely recognized as a valuable research contribution [2,6]. Similarly, the general CoT method [9], which can be viewed as a simple input augmentation approach via few-shot learning rather than introducing new model architectures, has demonstrated substantial performance improvements. StructCoT aligns with this paradigm by employing structural information in the reasoning process, enriching the input to improve downstream performance.
>
> While our primary focus is not on proposing a new model architecture, StructCoT introduces a significant contribution: the ability to assess the alignment between the generated structural rationale and the corresponding molecule. This enables a novel matching ratio-based rejection sampling mechanism, which ensures consistency between the rationale and the output. To the best of our knowledge, this is the first approach to evaluate the coherence between generated rationales and final outputs in molecular reasoning tasks.
>
>
> ---
>
> **W5. Incorporating more advanced concepts into the structural information may improve the method.**
>
> Thank you for your insightful suggestions. In our work, we selected six "structural" features of molecules as they are highly relevant to molecular properties, as illustrated in Figure 3. Nevertheless, to address your comments, we conducted additional experiments incorporating three advanced components: fingerprint information (Morgan fingerprint) and electronic properties (topological polar surface area (TPSA) and molar refractivity (MR)). Specifically, the Morgan fingerprint encodes local substructures within a specified radius, TPSA represents the sum of the surface areas of all polar atoms and their attached hydrogen atoms, and MR quantifies the total polarizability of a molecule.
>
> To verify the effectiveness of each additional descriptor, we evaluated the performance of molecule captioning using ChemT5-small. We provide the results in Figure 12 of Appendix B.3. We observed that incorporating all three additional descriptors together did not further improve the performance of StructCoT, although applying each additional descriptor individually improved performance. This validates the importance of structural information and the sufficiency of our proposed structural components.
>
> ---
>
> **Minor. The word choice of CoT is misleading as the reasoning is generated by LLM.**
>
> First, we clarify that StructCoT is generated by the LLM during the text-based molecule generation task, which aligns with the general meaning of CoT as intermediate reasoning steps generated by the model.
>
> In addition, our use of the term "CoT" follows prior works [3,5,6], where the CoT process has been extended to include reasoning enhanced by external tools. For instance,
> - ChemCrow [3] employed external tools such as literature search and SMILES to price to generate proper reasoning for LLM.
> - MultiTool-CoT [5] proposed to incorporate multiple external tools such as a calculator and knowledge retrieval during the reasoning process.
> - PoT [6] introduced to incorporate a program interpreter into the CoT generation process.
>
> These works demonstrate that incorporating external tools into the reasoning process does not conflict with the CoT framework. However, if you have alternative suggestions for terminology that better aligns with the context, we are open to adopting them.
>
>
> ---
>
> **Minor. Phrases like "We propose to" in the contributions section could imply a suggestion rather than a concrete result.**
>
> Thank you for the suggestion. We updated our manuscript with the sentence starting with "We design~".

---

> ### Author Response · Authors · 2024-11-22
>
> **Q3. Although the method improves molecule captioning performance, the real-world applications of these gains are unclear. Are there specific downstream tasks that would benefit directly from enhanced captioning? This is a question regarding the general line of research, and not specifically pointed at this paper.**
>
>
> Molecule captioning is a widely used method to evaluate the models' understanding of molecules [1,2,8]. Enhanced molecule captioning complements precise and detailed descriptions of molecules, which facilitate the understanding of molecules.
>
> For instance, improved captioning can enhance the interpretability of molecule generation models by providing human-readable rationales for generated structures. This can be particularly beneficial in tasks like virtual screening, where researchers need to quickly assess large libraries of molecules for potential candidates. Additionally, detailed molecular captions can assist in the training and evaluation of predictive models for bioactivity, toxicity, or other chemical properties, further extending the utility of the method. Thus, while molecule captioning might appear as a non-practical task, its implications are broad and impactful.
>
> ---
>
> [1] Li, Sihang, et al. "Towards 3d molecule-text interpretation in language models." ICLR 2024.
>
> [2] Fang, Yin, et al. "Mol-instructions: A large-scale biomolecular instruction dataset for large language models." ICLR 2024.
>
> [3] Bran, A. M., et al. "ChemCrow: Augmenting large-language models with chemistry tools." Nature Machine Intelligence 2024.
>
> [4] Pei, Q., et al. "BioT5: Enriching cross-modal integration in biology with chemical knowledge and natural language associations." EMNLP 2023.
>
> [5] Inaba, T., et al. "MultiTool-CoT: GPT-3 can use multiple external tools with chain of thought prompting." ACL 2023.
>
> [6] Xenhu Chen, et al., "Program of Thoughts Prompting: Disentangling Computation from Reasoning for Numerical Reasoning Tasks." TMLR 2023.
>
> [7] Ye, F., et al. "ProteinBench: A Holistic Evaluation of Protein Foundation Models." arXiv preprint 2024.
> [8] Edwards, C., et al. "Translation between molecules and natural language." ACL 2022.
> [9] Wei, J., et al. "Chain-of-Thought Prompting Elicits Reasoning in Large Language Models." NeurIPS 2022.

---

> ### Comment · Reviewer_NySt · 2024-11-25
> **Updated score**
>
> Thanks for your rebuttal. I have updated my score based on the additional experiments and clarifications.

---

> > ### Author Response · Authors · 2024-12-02
> >
> > Thank you for your insightful comments, which have significantly enhanced our paper. We sincerely appreciate your valuable feedback!

---

### Author Response · Authors · 2024-11-28
**General response**

Dear Area Chair and reviewers (**NySt**, **ZbGe**, **gWgH**, **GaNM**, and **atZJ**),

We would like to express our sincere gratitude for your time and efforts in reviewing our paper. Below, we summarize the concerns raised by the reviewers and detail how we have addressed them during the rebuttal phase.

- Reviewers **NySt** and **atZJ** were concerned about the lack of baselines in our experiments. We have conducted additional experiments and incorporated the results of integrating StructCoT with a recent baseline [1] into our revised manuscript. We believe this addition satisfactorily addresses these concerns.
- Reviewers **NySt**, **ZbGe**, **gWgH**, **GaNM** requested additional structural information and an ablation study for the proposed structural components. In response, we conducted experiments incorporating three advanced components: fingerprint information (Morgan fingerprint) and electronic properties (topological polar surface area (TPSA) and molar refractivity (MR)). Furthermore, we performed an ablation study on each structural component to evaluate their individual contributions.
- Reviewer **gWgH** requested additional experiments for different tasks. We have conducted experiments on retrosynthesis and verified that StructCoT consistently improves performance across tasks.
- Reviewers **NySt** and **atZJ** raised concerns about the usage of the term "CoT". We clarified that the term "CoT" aligns with prior works [2,3,4], where the CoT process has been extended to include reasoning enhanced by external tools.

Finally, we summarize the revisions of our updated manuscript.

- Additional experimental results on Mol-Instructions [1]
- Additional experimental results on the retrosynthesis task
- Additional experimental results on incorporating enhanced structural information
- Ablation study on matching ratio-based rejection sampling
- Ablation study on each structural component
- Clarification on the meaning of the comparison to ChemCrow
- Incorporation of all the editorial comments
All revisions have been highlighted in \textcolor{magenta}{magenta} in the updated manuscript for easy reference.

In conclusion, we beleive that we have thoroughly addresed the concerns of the reviewers through our comprehensive additional experiments, detailed responses and updates to the manuscript.

We appreciate your valuable time and effort,

Authors

[1] Fang, Yin, et al. "Mol-instructions: A large-scale biomolecular instruction dataset for large language models." ICLR 2024.

[2] Bran, A. M., et al. "ChemCrow: Augmenting large-language models with chemistry tools." Nature Machine Intelligence 2024.

[3] Inaba, T., et al. "MultiTool-CoT: GPT-3 can use multiple external tools with chain of thought prompting." ACL 2023.

[4] Xenhu Chen, et al., "Program of Thoughts Prompting: Disentangling Computation from Reasoning for Numerical Reasoning Tasks." TMLR 2023.

---

### Meta-Review · Area_Chair_VLDf · 2024-12-19

**Metareview:**

(a). The paper introduces StructCoT, a framework designed to enhance LLMs for molecular reasoning tasks by incorporating structural features.

(b).  The paper addresses the challenge of adapting LLMs to molecular reasoning, demonstrates practical performance improvements in molecule captioning and text-based molecule generation.

(c)  Lack of Novelty: The method relies on existing tools for input augmentation, offering limited innovation.  Limited Evaluation: Focuses on basic tasks like molecule captioning and ignores more complex applications and other strong baselines.

(d) The paper demonstrates incremental improvements in LLM-based molecular reasoning but lacks sufficient novelty, broader evaluation, and robust baseline comparisons. The reliance on pre-existing tools for structural information limits the technical contribution. Additionally, some reviewer concerns regarding the fundamental approach and clarity were not adequately addressed in the rebuttal. Given these factors, I recommend rejection.

**Additional Comments On Reviewer Discussion:**

The rebuttal phase addressed several reviewer concerns but did not resolve all  criticisms:

1. Baselines: While additional experiments were conducted with Mol-Instructions, more experiments with fine-tuning BioT5  and other state-of-the-art models were not included, citing computational limitations.

2. Novelty: Reviewers raised concerns about the lack of technical innovation, and the rebuttal’s clarification about aligning StructCoT with CoT paradigms was not convincing to all reviewers.

3. Presentation: Minor issues, such as figure labels, were addressed, but they did not significantly impact the overall assessment.

In weighing these points, the inability to address core concerns about novelty, evaluation breadth, and baselines justified the decision to reject.

---

### Decision · Program_Chairs · 2025-01-22

Reject